# Statistical characteristics of analytical studies published in Peruvian medical journals from 2021 to 2022: A methodological study

Natalia Nombera-Aznaran[1], David Guevara-Lazo[1], Daniel Fernandez-Guzman[2], Alvaro Taype-Rondán[3,4]*

1 Facultad de Medicina Humana, Universidad Peruana Cayetano Heredia, Lima, Perú, 2 Carrera de Medicina Humana, Universidad Científica del Sur, Lima, Perú, 3 Unidad de Investigación para la Generación y Síntesis de Evidencias en Salud, Universidad San Ignacio de Loyola, Lima, Perú, 4 EviSalud—Evidencias en Salud, Lima, Perú

* alvaro.taype.r@gmail.com

## Abstract

**Data Availability Statement:** All relevant data are within the paper and its Supporting Information files.

### Objective

While statistical analysis plays a crucial role in medical science, some published studies might have utilized suboptimal analysis methods, potentially undermining the credibility of their findings. Critically appraising analytical approaches can help elevate the standard of evidence and ensure clinicians and other stakeholders have trustworthy results on which to base decisions. The aim of the present study was to examine the statistical characteristics of original articles published in Peruvian medical journals in 2021–2022.

### Design and setting

We performed a methodological study of articles published between 2021 and 2022 from nine medical journals indexed in SciELO-Peru, Scopus, and Medline. We included original articles that conducted analytical analyses (i.e., association between variables). The statistical variables assessed were: statistical software used for analysis, sample size, and statistical methods employed (measures of effect), controlling for confounders, and the method employed for confounder control or epidemiological approaches.

### Results

We included 313 articles (ranging from 11 to 77 across journals), of which 67.7% were cross-sectional studies. While 90.7% of articles specified the statistical software used, 78.3% omitted details on sample size calculation. Descriptive and bivariate statistics were commonly employed, whereas measures of association were less common. Only 13.4% of articles (ranging from 0% to 39% across journals) presented measures of effect controlling for confounding and explained the criteria for selecting such confounders.

**Funding:** The author(s) received no specific funding for this work.

## Conclusion

This study revealed important statistical deficiencies within analytical studies published in Peruvian journals, including inadequate reporting of sample sizes, absence of measures of association and confounding control, and suboptimal explanations regarding the methodologies employed for adjusted analyses. These findings highlight the need for better statistical reporting and researcher-editor collaboration to improve the quality of research production and dissemination in Peruvian journals.

## Introduction

Health research plays a crucial role in understanding the afflictions that impact humanity and in making informed healthcare decisions [1]. However, when research fails to adhere to best practices in methodology and statistics, the credibility of its results may be compromised. This can lead to conclusions that lack logic and fail to establish real associations, hindering the implementation of health policies. While the primary responsibility lies with the study authors, the efforts of reviewers and journal editors play a vital role in refining scientific articles before their publication [2].

Given the importance of ensuring adequate statistical methodologies, previous studies have examined this subject within medical journals. A study performed in the Journal of the American Medical Association (JAMA) showed that in 2010 around half of papers used advanced adjusted regression models [3]. Conversely, a study in a Colombian medical journal highlighted substandard reporting of statistical regression models [4]. Moreover, an examination of statistical methodologies in orthopedic papers revealed that one-third needed an alternative analysis [5]. In 17%, choosing a different statistical analysis could have changed the conclusions.

While statistical methodology holds universal importance in medical research, it poses specific challenges in emerging research contexts. Countries that are in the process of developing their healthcare research capacities require local researchers to best understand existing gaps and priorities for improvement. Peru has witnessed a surge in research activity in recent decades [6, 7], yet there are opportunities to enhance methodological rigor. Analyzing statistical practices in Peruvian medical journals can offer insights tailored to the country's needs, considering the ongoing development of both the educational system and publication culture. Identifying areas of strength in statistical analysis, as well as aspects that require strengthening, will help Peruvian researchers, reviewers, and editors, as well as those in similar contexts where local research is on the rise, to elevate the standard of evidence. This ensures that clinicians and other stakeholders have reliable results upon which to base their decisions. Hence, the objective of this study was to analyze the statistical characteristics of original analytical articles published in Peruvian medical journals.

## Methods

### Design and data source

We conducted a methodological study encompassing all original analytical articles published in Peruvian medical journals indexed in SciELO-Peru, Scopus, or Medline, between 2021 and 2022.

We included all medical journals that were indexed in SciELO-Peru, Scopus, or Medline during data extraction (April 2023). These were nine: "Acta Médica Peruana", "Revista Médica

Herediana", "Revista de la Facultad de Medicina Humana", "Horizonte médico", "Revista del Cuerpo Médico del Hospital Nacional Almanzor Aguinaga Asenjo", "Revista de Neuro-Psiquiatría", "Revista Peruana de Ginecología y Obstetricia", "Revista de Gastroenterología del Perú", and "Revista Peruana de Medicina Experimental y Salud Pública".

Subsequently, we examined the official websites of these journals to scrutinize every article published in their volumes corresponding to the years 2021 and 2022. Our inclusion criteria encompassed original studies on human subjects, incorporating analytical assessments (i.e., associations between variables using any bivariate association analyses). We excluded secondary studies such as systematic reviews or scoping reviews, as well as validation studies.

### Data collection process and variables

Two authors (NNA and DGL) independently screened the articles carefully to identify those who met the inclusion criteria. In cases of disagreement, a third author was required to resolve any discrepancy (DFG or ATR). Data extraction was performed into an Excel spreadsheet and was independently carried out by two of the authors (NNA and DGL).

The general variables extracted from each article included: journal name, year of publication, study design or field of study. This field was categorized into clinical or public health types. Clinical articles focus on diagnosing, treating, and managing patients individually or in groups. Public health articles address disease prevention, health promotion, policy, epidemiology, and more. Additional variables encompassed time between submission and acceptance of the manuscript, number of authors, and presence of any Peruvian authors.

The statistical variables assessed were: statistical software, sample size, and employed statistical methods. These include measures of effect and controlling for confounders. Measures of effect included linear regression beta coefficient [β], odds ratio [OR], prevalence ratio [PR], relative risk [RR], hazard ratio [HR]. Methods employed for cofounder control can be statistical or epidemiological approaches. Statistical methods rely on p-values or other statistical methods, while epidemiological approaches involve theoretical rationale and clinical relevance. These variables were chosen according to previous studies [4, 5, 8–10]. This data was gathered by thoroughly reviewing each article, with particular focus on the methods and results sections, although additional sections were consulted when required.

### Statistical analysis

Data entry was conducted using Microsoft Excel, and subsequent statistical analyses and graphics were carried out using R software, version 4.1.0 (R Foundation for Statistical Computing, Vienna, Austria). To provide a comprehensive overview of the data, descriptive statistics were employed. The results were presented in the form of either medians and interquartile ranges or means and standard deviations, depending on the normality distribution, which was assessed using the Shapiro-Wilk test. Categorical variables were summarized using counts and percentages. Bar graphs were generated to illustrate the frequency of the softwares, statistics, graphics and the calculation of effect measures in the included articles.

### Results

In the volumes corresponding to 2020 and 2021 of the nine Peruvian medical journals, a total of 1361 published articles (original and non-original articles) were initially identified. Subsequently, 564 original articles were evaluated. After applying inclusion and exclusion criteria, 313 studies (150 in 2021 and 163 in 2022), which have incorporated analytical assessments, were included. Of these, 69.6% focused on clinical research and 30.4% in public health. Regarding the study design, cross-sectional studies were the most common, comprising 67%

of the total, followed by prospective cohorts (14.7%) and case-control studies (8%), with clinical trials (1.3%) and ecological studies (1%) being the least prevalent. Concerning the authorship, the median number of authors were 4, and 83.1% of the articles had at least one author affiliated with an institution in Peru. As for the publication process, the median time between submission and acceptance of articles was 92.5 days (Table 1).

We found different patterns regarding the statistical analysis in the methodology section of the included articles. The majority of the studies (90.1%) specified the statistical software used in their analyses. There was a preference for SPSS (42.5%) and Stata (38%), and less commonly used softwares were R (3.2%), Epidat (1.6%), EpiInfo (1.3%), etc (Fig 1A). Relating to garding the sample size calculation, we found that 78.3% of the studies did not mention their sample size or statistical power calculation, whereas only 18.2% specified their sample size calculation. Among the studies that calculated sample size (n = 57), 77% performed calculations based on the study design, but 56.1% did not report the use of parameters based on previous studies. Also, 61.4% did not report the statistical power used in the calculations (Table 2).

For analysis methods, we included studies that performed any bivariate association analyses (hypothesis tests), yet merely 42.5% included association analyses with regressions or

**Table 1. General characteristics of analytical original studies published in Peruvian indexed in SciELO-Peru, 2021–2022 (n = 313).**

| Characteristics | Total n (%) |
|---|---|
| **Journal** | |
| Revista de la Facultad de Medicina Humana | 77 (24.6) |
| Revista del Cuerpo Médico del Hospital Nacional Almanzor Aguinaga Asenjo | 60 (19.2) |
| Horizonte Médico | 46 (14.7) |
| Revista Peruana de Medicina Experimental y Salud Pública | 41 (13.1) |
| Acta Médica Peruana | 26 (8.31) |
| Revista de Gastroenterología del Perú | 20 (6.4) |
| Revista Peruana de Ginecología y Obstetricia | 17 (5.4) |
| Revista Médica Herediana | 15 (4.8) |
| Revista de Neuro-Psiquiatría | 11 (3.5) |
| **Year of publication** | |
| 2021 | 150 (47.9) |
| 2022 | 163 (52.1) |
| **Study area** | |
| Clinical | 218 (69.6) |
| Public health | 95 (30.4) |
| **Study design** | |
| Cross-sectional | 212 (67.7) |
| Prospective cohort | 46 (14.7) |
| Case control | 25 (8.0) |
| Quasi-experimental | 13 (4.2) |
| Retrospective cohort | 10 (3.2) |
| Randomized clinical trial | 4 (1.3) |
| Ecological | 3 (1.0) |
| **Time between submission and acceptance, in days**[a] | 92.5 (50.5–146.3) |
| **Number of authors**[a] | 4 (3–6) |
| **Had at least one author with Peruvian affiliation** | 260 (83.1) |

[a] Median (percentile 25 –percentile 75)

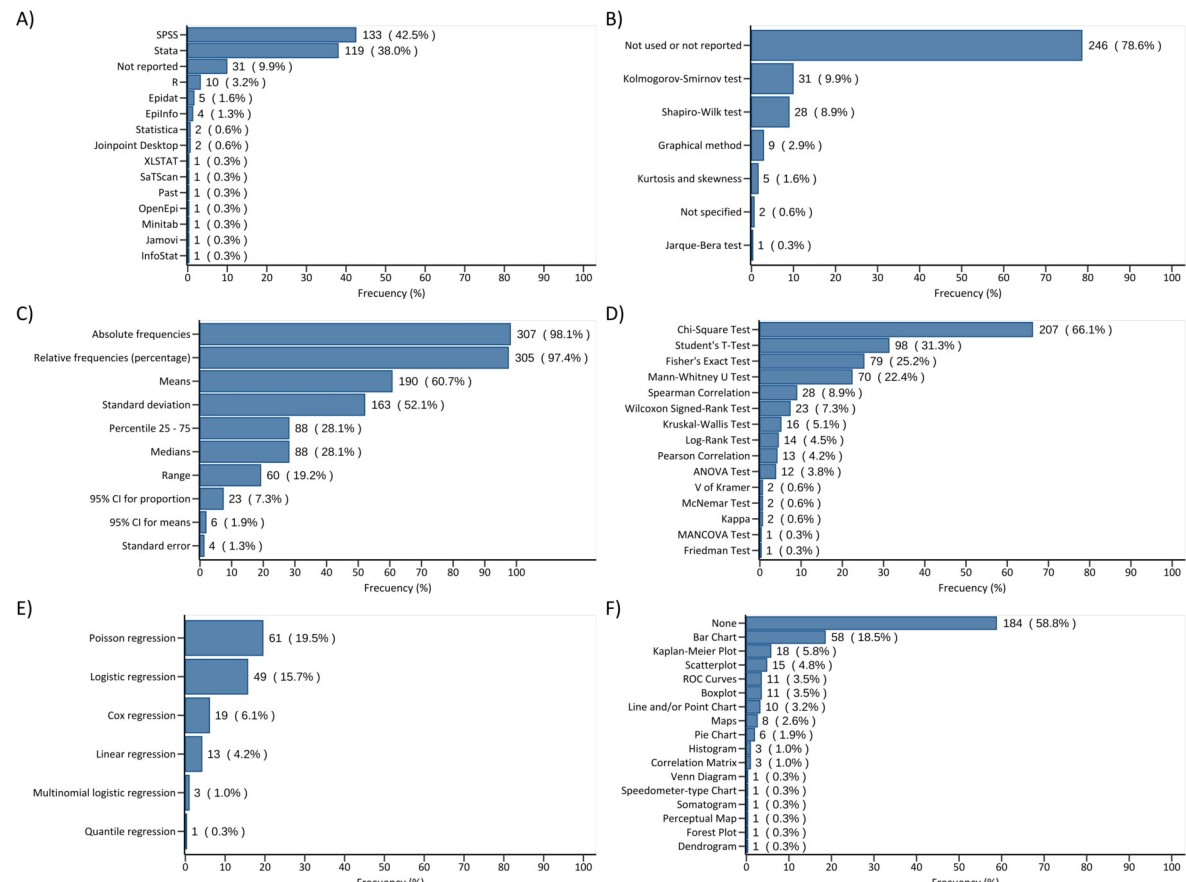

**Fig 1. Panel on software, statistics, and graphics used in the included articles.** A) Software (categories were not mutually exclusive); B) Normality assessment methods; C) Descriptive statistics; D) Bivariate statistics (hypothesis tests); E) Regression models; F) Graphics.

generalized linear models (Fig 1E). Other types of statistical analyses including complex survey sampling (3.5%), diagnostic test analysis (3.5%), time series analysis (0.3%), geospatial analysis (0.3%), and machine learning analysis (0.6%) were infrequent (Table 2). Furthermore, 47.6% did not present measures of effect, 23.03% only presented crude measures of effect, 15.7% presented adjusted measures of effect but did not explain the criteria used for selecting confounders; and 13.4% presented adjusted measures of effect and explained the criteria used for selecting confounders (Table 2). Regarding those criteria, 57.1% employed a statistical approach, though 45.2% utilized an epidemiological approach. Most studies (78.6%) did not include or did not report how the normality assessment of numerical variables was performed. On the other hand, those that employed normality assessment used the Kolmogorov-Smirnov (9.9%) and Shapiro-Wilk (8.9%) tests followed by the graphical method (2.9%). Additionally, two studies mentioned performing the normality assessment but did not specify the method used.

On the subject of the bivariate analysis, the Chi-Square test was the most common (66.1%), followed by Student's t-test (31.3%), Fisher's test (25.2%), and the Mann-Whitney U test (22.4%). Less frequently utilized tests included the Log-Rank test (4.5%), Pearson correlation (4.2%), and ANOVA test (3.8%). Regarding regression models, Poisson regression (19.5%) and logistic regression (15.7%) emerged as the most commonly employed, while linear regression (4.2%) and Multinomial logistic regression (1%) were less common. Other analyses, such

**Table 2. Features of the reported methodology on statistical analyses conducted in the included studies (n = 313).**

| Feature | Total |
|---|---|
| | N (%) |
| **Specifies the statistical software used for the analyses** | 284 (90.7) |
| **Mentions how the sample size calculation or statistical power was performed** | |
| None of them | 245 (78.3) |
| Sample size | 57 (18.2) |
| Statistical power | 11 (3.51) |
| **Analysis performed[a]** | |
| Present bivariate association analyses | 313 (100.0) |
| Present descriptive analyses | 311 (99.4) |
| Present association analyses with regressions or generalized linear models | 133 (42.5) |
| Present analyses for surveys with complex sampling (e.g., ENDES) | 11 (3.5) |
| Present diagnostic test analyses | 11 (3.5) |
| Present analyses with Machine Learning | 2 (0.6) |
| Present time series analyses | 1 (0.3) |
| Present geospatial analyses | 1 (0.3) |
| Perform imputation of missing data | 0 (0.0) |
| **In tables/figures that present p-values, is it explicitly stated which test was used to calculate p-values?** | |
| Do not provide p-value in tables | 28 (9.0) |
| No, provide p-value, but do not specify the test | 152 (48.6) |
| Yes, but not for all tests | 36 (11.5) |
| Yes, all tables/figures specify the test | 97 (31.0) |
| **Present measures of effect?** | |
| Do not present measures of effect | 149 (47.6) |
| Only present crude measures of effect | 73 (23.3) |
| Present adjusted measures of effect, but do not explain the criteria used to select confounders[b] | 49 (15.7) |
| Present adjusted measures of effect and explain the criteria used to select confounders[b] | 42 (13.4) |
| **Measures of effect used (n = 164 studies that present measures of effect)** | |
| Odds Ratio (OR) | 66 (40.2) |
| Prevalence Ratio (PR) | 58 (35.4) |
| Risk Ratio (RR) | 19 (11.6) |
| Hazard Ratio (HR) | 13 (7.9) |
| Beta coefficient | 6 (3.7) |
| Prevalence Ratio (PR) and Risk Ratio (RR) | 1 (0.6) |
| Hazard Ratio (HR) and Odds Ratio (OR) | 1 (0.6) |
| **Use a statistical approach for selecting variables in the adjusted model (n = 91 studies that explained this criteria)** | |
| Do not specify a statistical or epidemiological approach for selecting variables | 67 (73.6) |
| P-value < 0.05 in the bivariate model | 15 (16.5) |
| P-value < 0.20 in the bivariate model | 6 (6.6) |
| Information criteria (Akaike or Bayesian) | 2 (2.2) |
| Other criteria | 1 (1.1) |

*(Continued)*

**Table 2.** (Continued)

| Feature | Total |
|---|---|
| | N (%) |
| **Use an epidemiological approach for selecting variables in the adjusted model (n = 91 studies that explained this criteria)** | 19 (20.9) |

[a]Since categories were not mutually exclusive, the total percentage can exceed 100%.

[b]All studies that controlled for confounders did it using adjusted regressions.

Abbreviations: ENDES: National Demographic and Family Health Survey; OR: Odds Ratio; PR: Prevalence Ratio; RR: Relative Risk; HR: Hazard Ratio.

as Machine Learning (0.6%), time series analysis (0.3%), and geospatial analysis (0.3%), were performed even less frequently. Most studies (58.8%) lacked graphical representations; bar charts and Kaplan-Meier plots were most common, as depicted in Fig 1F.

Across the included journals, the percentage of articles lacking effect measures varied from 28.3% to 86.7%, the percentage of articles that displayed adjusted measures varied from 0% to 46.3%, and the percentage of articles that presented adjusted measures with an explanation of confounder selection criteria varied from 0% to 39.0%. The Revista Peruana de Medicina Experimental y Salud Pública stands out with the highest percentage of original articles reporting adjusted effect measures and explaining the criteria used for confounder selection (39%), followed by the Revista del Cuerpo Médico del Hospital Nacional Almanzor Aguinaga Asenjo (21.7%). In contrast, the majority of articles in the Revista Médica Herediana (86.7%) did not employ measures of effect (86.7%). All these results are displayed in Fig 2.

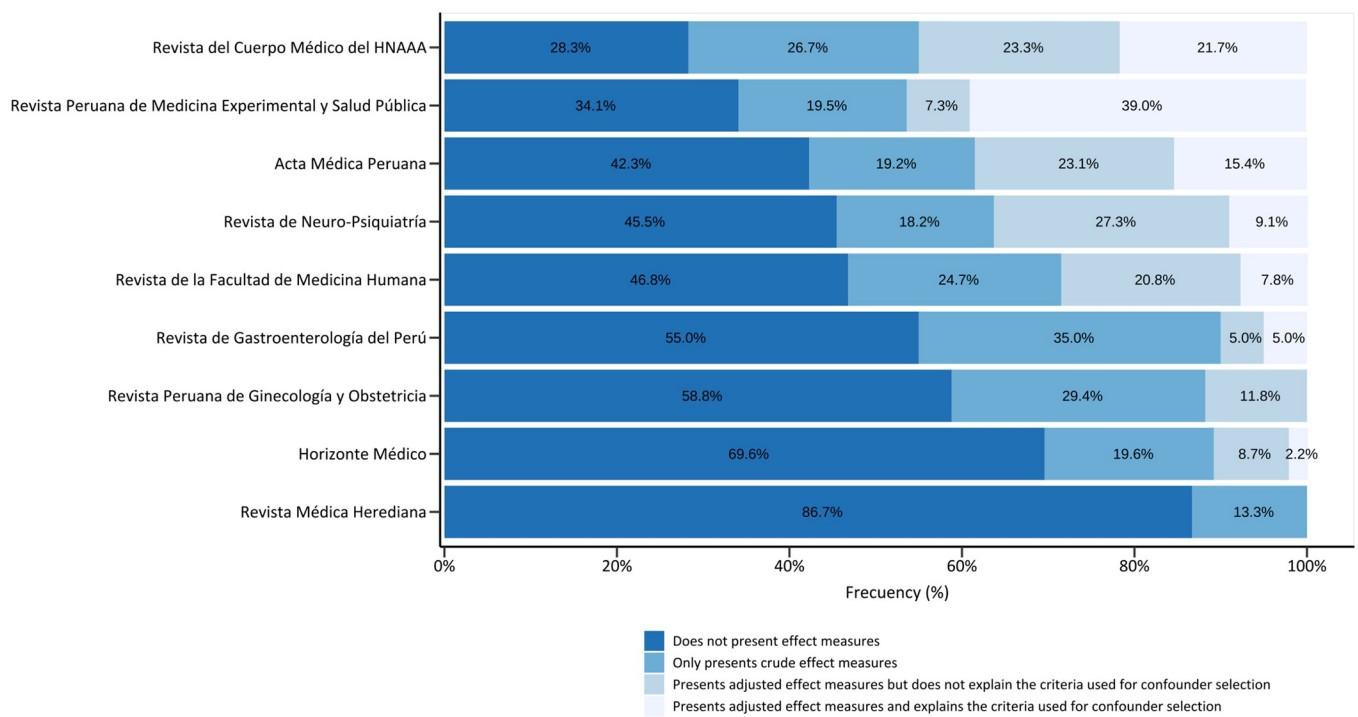

**Fig 2. Calculation of effect measures in the included articles, per journal (N = 313).** Reporting effect measures, including prevalence ratio, relative risk, odds ratio, etc., is increasingly important. They allow the translation of research findings into clinically relevant insights that can be useful for evidence-based care.

## Discussion

### Sample size and statistical power

Whether the sample is collected for the study or obtained from a pre-existing database, determining the sample size is crucial. This ensures to obtain the minimum number of subjects necessary to detect specific associations. Similarly, in secondary data analysis, assessing whether non-significant findings are due to a lack of statistical power is important.

However, a significant proportion of original articles (78%) show deficiencies in specifying the calculation of sample size or statistical power. Sample size addressing may omit essential computation details, hindering replicability despite its acknowledgment in the analysis. Kumar et al. reported similar results examining statistics application trends in 300 biomedical research articles. They found that, despite the mention of sample size in 95% of the articles, reproducibility was limited to only 7%. In the remaining articles, neither the precise mention of the sample size estimation technique nor an adequate description was provided [11]. Likewise, in 448 Balkan Medical Journal studies, sample size reporting was 8.1% in cross-sectional studies, 13.8% in cohorts, and 16% in clinical trials [12]. Even in high-impact general medical journals, only one-third of articles adequately describe sample size calculations [13].

### Measures of effect

Our study found that 47% of articles lacked measures of effect. A parallel trend was identified in a study that assessed 60 observational studies in high-impact occupational medicine and health journals [14]. Although all studies reported outcome events or summary measures, only 60% provided unadjusted estimates, and merely 43.3% included confounder-adjusted estimates [14]. Similarly, an examination of JAMA articles between 1990 and 2010 uncovered an increase in incorporating epidemiologic statistics, reaching almost 30% [3]. This highlights a gap in reporting standards, emphasizing the need for improved consistency in presenting both unadjusted and adjusted estimates. In contrast, an analysis of 113 studies from a Colombian journal (Biomedica, 2000–2017) revealed nearly 90% reported measures of association coefficients [4].

Measuring the strength of risk factor-outcome associations is an important goal in epidemiological and clinical research [15]. Effect sizes—both unstandardized and standardized from individual studies—should be reported for inclusion in systematic reviews and meta-analyses [16]. However, our analysis revealed that only 13.4% of articles provided criteria for selecting confounding variables, essential to adjusted analyses. These can be chosen based on established causal relationships, statistical associations (p-value < 0.05), or an epidemiological approach. In our study, confounding variables were primarily selected through the last two methods.

Adequate management of confounding variables is essential for isolating the true impact of the exposure under investigation (causal inference) [17–20]. Specifying criteria for selecting confounding variables in an adjusted model analysis ensures systematic identification and control, enhancing research reliability. Unclear descriptions of statistical techniques addressing confounding factors and biases hinder reader assessment of result validity.

Despite explicit reporting recommendations for handling missing data within the STrengthening the Reporting of OBservational studies in Epidemiology (STROBE) guidelines [21], we found no articles adhering to these recommendations. Failing to handle missing data appropriately may lead to incorrect conclusions and potential harm to patients if clinical decisions are based on incomplete or biased information [16].

Even among high-impact medical journals, deficiencies persist in reporting STROBE items. For instance, high-impact occupational medicine and health journals reported all STROBE

items in only 63% of articles, while otorhinolaryngology journals adequately reported a mean of 51.4% of STROBE items [14, 22]. Our findings underscore the need for enhanced reproducibility and quality reporting in observational studies published in Peruvian journals. Therefore, addressing this issue should be a central point for both authors and editors when reporting or reviewing observational studies.

In scientific research, there are three main approaches: describing, explaining and predicting [23]. Describing refers to characterizing and summarizing data without attempting to establish causal relationships, focusing on the who, what, when and where of the phenomena studied. Explaining goes beyond description, seeking to understand the why and how of the phenomena, which implies identifying causal relationships and underlying mechanisms. To this end, it is crucial to adequately control for confounding variables and biases. Prediction, on the other hand, focuses on the use of data and models to anticipate future events or behaviors, based on patterns identified in historical data. This approach requires robust models that are able to generalize to new data and situations.

Despite the extensive use of statistical analysis in the studies reviewed, no research was found that adequately assessed causality. Many of the articles were limited to describing associations using bivariate analyses and regression models, without comprehensively addressing the criteria necessary to infer causal relationships. This includes control for selection bias and adequate control for confounding variables, as well as insufficient explanation of the methods used for this selection [24]. This limitation suggests that, although different statistical techniques were applied, they were not used with the necessary rigor to establish causality, which underlines that many of the studies have a descriptive approach.

## Inadequate presentation of the data

Our analysis identified a frequent oversight in the presentation of data within the examined articles. The mention of the statistical software was absent in 9.9% of the articles. Furthermore, 48.6% of the articles presented tables/figures that included p-values but in which the statistical tests used were not specified.

Tables play a crucial role in presenting research results. Even when presented independently, they should include essential elements necessary for communicating information, such as population parameters, alongside confidence intervals, p-values, and the utilized test statistics [25]. Inclusion of these components in tables facilitates interpretation of results for readers. The presentation of p-values and confidence intervals enables clinicians to discern the statistical significance and understand the differences in outcomes between two groups [26]. While p-values indicate statistical significance, confidence intervals establish the threshold for the magnitude of the effect. Both of these statistical measures are integral in the interpretation of clinical outcomes. Therefore, ensuring their inclusion in tables enhances the clarity and meaningfulness of research findings.

## Peruvian journals

Peru encounters significant challenges regarding the representation of its journals in key international databases like Scopus and Medline. At the time of the study, only two Peruvian journals, "Revista Peruana de Medicina Experimental y Salud Pública" and "Revista de Gastroenterología del Perú," are indexed in both Medline and Scopus. This aligns with the current national situation, where we have approximately 92 health science journals, with around 95% remaining unindexed [27].

This scenario is significantly impacted by crucial factors, stemming from a shortage of researchers and a prevalent inclination to publish their research in outside journals indexed in

international databases such as Scopus or Medline [28]. Additional challenges include insufficient financial support and promotion for national medical journals. Collectively, these factors diminish visibility and recognition both regionally and internationally.

Our study highlights significant concerns regarding the methodological reporting of statistical processes in articles published in Peruvian journals. Introducing a statistical review of the submitted articles as an integral part of the editorial process, as suggested in our study, could be a replicable strategy for enhancing the reporting quality of research articles in diverse international contexts and improving confidence in their results [29].

National journals are important to spread studies of local interest, which may be of special interest for local practitioners and stakeholders [30]. Nevertheless, as revealed by our study, studies published in local Peruvian medical journals need to improve their statistical report, which ultimately may underpin their credibility and usage.

Efforts to identify and improve the statistical deficiencies of Peruvian medical journals can foster a more inclusive and comprehensive representation of diverse perspectives in the global scientific community. By recognizing the challenges faced by Peruvian journals and proposing solutions, our study aims to contribute to a broader conversation on strengthening the integrity and impact of research from developing regions, ultimately fostering a more interconnected and equitable landscape for scientific knowledge dissemination.

## Limitations and strengths

The findings of this study shed light on critical aspects of statistical reporting in Peruvian biomedical articles. However, some limitations should be taken into account when interpreting the results. Firstly, the temporal scope of the study, which focused on publications from 2021 and 2022 to reflect the current state of the topic, but may not be generalizable to different time periods, as statistical reporting may change over an extended time. Additionally, while the study reviewed statistical reporting, it did not thoroughly cover all aspects of research methodology.

It is essential to recognize that despite the limitations reported in the medical Peruvian journals, they do not fully represent the medical research produced in Peru, since the majority of Peruvian biomedical articles are published in international journals, which were not the focus of this investigation [6, 31]. Analyses of Peruvian research revealed that only 11.3% of COVID articles up to 2021 [31] and just over 19.1% of public health articles up to 2020 [6] were published in Peruvian journals. However, it is relevant to note that the primary objective of our study was to assess Peruvian journals to provide insights for their improvement.

Turning to the strengths of the study, to our knowledge, this is the first study conducted in Peru and within Spanish journals assessing statistical reporting in medical literature. Furthermore, a thorough article analysis was carried out in duplicate to ensure optimal data extraction.

## Conclusion

Our study reveals significant weakness in analytical studies published in Peruvian journals including inadequate reporting of sample size, absence of measures of association and adjusted analyses, suboptimal documentation of the methods used for adjusted analyses, and inadequate presentation of the data. These findings underscore the importance of reinforcing reporting statistical standards in biomedical research. Moreover, the study highlights the importance for collaborative efforts between the researchers and the editorial team. By enhancing these aspects, Peruvian journals can improve research quality and promote global dissemination, particularly focusing on conditions unique to Peruvian populations.

## Supporting information

**S1 Dataset. Dataset containing variables extracted from original articles published in Peruvian journals.**
(XLSX)

## Author Contributions

**Conceptualization:** Alvaro Taype-Rondán.

**Data curation:** Natalia Nombera-Aznaran, David Guevara-Lazo.

**Formal analysis:** Daniel Fernandez-Guzman.

**Writing – original draft:** Natalia Nombera-Aznaran, David Guevara-Lazo, Daniel Fernandez-Guzman, Alvaro Taype-Rondán.

**Writing – review & editing:** Natalia Nombera-Aznaran, David Guevara-Lazo, Daniel Fernandez-Guzman, Alvaro Taype-Rondán.

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
