## [Decision Letter · Decision Letter 0]

29 May 2024

PONE-D-24-15400Statistical characteristics of analytical studies published in Peruvian medical journals from 2021 to 2022: a methodological studyPLOS ONE

Dear Dr. Taype-Rondan,

Thank you for submitting your manuscript to PLOS ONE. After careful consideration, we feel that it has merit but does not fully meet PLOS ONE’s publication criteria as it currently stands. There are a few additions and corrections that are required, which does not merit a rejection. Therefore, we invite you to submit a revised version of the manuscript that addresses the points raised during the review process.

I have also included a file with a few comments, for your review.

We look forward to receiving your revised manuscript.

Kind regards,

Alfredo Luis Fort, M.D., M.Sc., Ph.D.

Academic Editor

PLOS ONE

Journal Requirements:

3. Please remove your figures from within your manuscript file, leaving only the individual TIFF/EPS image files, uploaded separately. These will be automatically included in the reviewers’ PDF.

Reviewers' comments:

Reviewer's Responses to Questions

**Comments to the Author**

1. Is the manuscript technically sound, and do the data support the conclusions?

Reviewer #1: Yes

Reviewer #2: No

Reviewer #3: Partly

2. Has the statistical analysis been performed appropriately and rigorously? 

Reviewer #1: Yes

Reviewer #2: No

Reviewer #3: Yes

3. Have the authors made all data underlying the findings in their manuscript fully available?

Reviewer #1: Yes

Reviewer #2: No

Reviewer #3: Yes

4. Is the manuscript presented in an intelligible fashion and written in standard English?

Reviewer #1: Yes

Reviewer #2: No

Reviewer #3: Yes

5. Review Comments to the Author

Reviewer #1: The article makes significant contributions to the improvement of medical research in Peruvian journals, however, it would have been interesting to expand the time limit beyond 2021-2022 to give readers a broad scope of the extent of the research objective - statistical analytic strength of medical research in Peruvian journals.

Reviewer #2: As the authors say in the abstract," some published

studies might have utilized suboptimal analysis methods, potentially undermining the

credibility of their findings. Critically appraising analytical approaches can help elevate the

standard of evidence and ensure clinicians and other stakeholders have trustworthy results on

which to base decisions."

I was expecting that they will suggested many optimal techniques mentioning the weaknesses of already used Statistical techniques. But I am disappointed authors just gave some descriptions mostly in the form of %age but they did not suggest better replacements.

Authors' statistical knowledge seems very limited, e.g., see the sentence" The statistical variables assessed were: statistical software used for analysis, sample size, and statistical methods employed (measures of effect), controlling for confounders, and the method employed for confounder control or epidemiological approaches." Are all these statistical variables?

I did not see any worth in the presented work.

Reviewer #3: The study provides a valuable contribution to understanding the state of statistical reporting in

Peruvian biomedical articles. Despite its limitations, the findings highlight critical areas for

improvement and set the stage for future research. By addressing the identified deficiencies and

implementing the recommended improvements, the quality and credibility of Peruvian biomedical

research can be significantly enhanced.

6. PLOS authors have the option to publish the peer review history of their article (what does this mean?). If published, this will include your full peer review and any attached files.

Reviewer #1: No

Reviewer #2: **Yes: **Tanvir Ahmad

Reviewer #3: No

---

## [Author Response · Author response to Decision Letter 0]

10 Jun 2024

Response to Reviewer Comments:

• Comment #1: In the abstract section, the keywords are not arranged in alphabetical order which needs to be rewritten

Response: Dear Reviewer, thank you for your suggestion. We arranged the keywords in alphabetical order and they were selected from the MESH database. 

• Comment #2: On page number 12 section, regarding the Data collection process and variables, the authors did not mention the (NNA, DGL, DFG, and ATR). 

Response: Dear Reviewer, thank you for your comment. We detailed the first paragraph regarding the data collection process and variables. In addition we mention the section each researcher was involved in. The section is: “Two authors (NNA and DGL) independently screened the articles carefully to identify those who met the inclusion criteria. In cases of disagreement, a third author was required to resolve any discrepancy (DFG or ATR). Data extraction was performed into an Excel spreadsheet and was independently carried out by two of the authors (NNA and DGL).”

• Comment #3: Page number 13, the section on statistical analysis is not attractive to the readers. The authors give very limited information for the analysis.

Response: Dear Reviewer, thank you for your comment. We improved and detailed the section of statistical analysis. It was redacted as follows: Data entry was conducted using Microsoft Excel, and subsequent statistical analyses and graphics were carried out using R software, version 4.1.0 (R Foundation for Statistical Computing, Vienna, Austria). To provide a comprehensive overview of the data, descriptive statistics were employed. The results were presented in the form of either medians and interquartile ranges or means and standard deviations, depending on the normality distribution, which was assessed using the Shapiro-Wilk test. Categorical variables were summarized using counts and percentages. Bar graphs were generated to illustrate the frequency of the softwares, statistics, graphics and the calculation of effect measures in the included articles.

• Comment #4: The explanations provided for the methodologies used in adjusted analyses were often insufficient. This includes unclear descriptions of the statistical techniques used to account for confounding factors and biases, making it difficult for readers to assess the robustness and validity of the results.

Response: Dear review, thank you for your comment. We have included a sentence addressing that aspect in our discussion: Unclear descriptions of statistical techniques addressing confounding factors and biases hinder reader assessment of result validity. 

• Comment #5: Page number 13, “We reported medians and 25th and 75th percentiles” The authors did not mention where these percentiles are given in any table or figure.

Response: Dear Reviewer, thank you for your comment. We clarified in Table 1 that the 25th and 75th percentiles for the time between submission and acceptance are 50.5–146.3 days, and for the number of authors are 3–6.

• Comment #6: The information given in the result section is very limited, this section needs to add more description.

Response: Dear Reviewer, thank you for your recommendation. We thoroughly expanded the results section to provide comprehensive coverage of our findings. We further describe the tables and figures along with their respective citations. 

• Comment #7: The length of sentences in this research article exceeded more than 20 words, which is not attractive to readers internationally.

Response: Dear Reviewer, thank you for your comment. We edited the text to ensure that each sentence is no longer than 20 words.

Response to In-Line Reviewer Comments:

• Comment 1: It may be of interest to add a "Recommendations" section, based on your findings?

Response: Dear Reviewer, thank you for your suggestion. We added a sentence of recommendation in the conclusion section in the abstract: These findings highlight the need for better statistical reporting and researcher-editor collaboration to improve the quality of research production in Peruvian journals.

• Comment 2: Not only credibility, but some results may be insufficiently deepened (e..g., no associations, regression, etc.), so that actual key final variables are found. If this does not happen, actual policies to improve healthcare may not be able to be implemented.

Response: Dear Reviewer, thank you for your input. We agree with your statement, which served as motivation for undertaking this research. As such, we have included a sentence addressing that aspect in our introduction: This can lead to conclusions that lack logic and fail to establish real associations, hindering the implementation of health policies.

• Comment 3: Any more information here...globally, etc.?

Dear Reviewer, thank you for your comment. Despite conducting a thorough bibliographic search, we were unable to find comprehensive information that addresses your concern.

• Comment 4: It would be good for general readers to explain a bit more what is meant by this.

Response: Dear Reviewer, thank you for your comment. We highlight the importance of reporting effect measures as you recommended in the section of results. “Reporting effect measures, including prevalence ratio, relative risk, odds ratio, etc., is increasingly important. They allow the translation of research findings into clinically relevant insights that can be useful for evidence-based care.”

• Comment 5: Good

Response: Dear reviewer, we appreciate your positive comment regarding the discussion of our article.

• Comment 6: "in" journals or "outside" journals? Unclear…

Response: Dear Reviewer, thank you for your comment. We accept your suggestion and include the term “outside” to further clarify our statement.

• Comment 7: Important to highlight. Do you have an actual figure for this "majority"? That would help with interpretation…

Response: Dear Reviewer, thank you for your comment. We provide two figures to address this aspect based on two bibliometric analyses of Peruvian research. Furthermore, we have amended our limitations section: Analyses of Peruvian research revealed that only 11.3% of COVID articles up to 2021 [31] and just over 19.1% of public health articles up to 2020 [6] were published in Peruvian journals.

Bibliography:

6. Sevillano-Jimenez J, Carrión-Chambilla M, Espinoza-Lecca E, Mori-Quispe E, Contreras-Pulache H, Moya-Salazar J. A bibliometric analysis of 47-years of research on public health in Peru. Electron J Gen Med. 2023 Jul 1;20(4):em488.

31. Vásquez-Uriarte K, Roque-Henriquez JC, Angulo-Bazán Y, Ninatanta Ortiz JA, Vásquez-Uriarte K, Roque-Henriquez JC, Angulo-Bazán Y, Ninatanta Ortiz JA. Análisis bibliométrico de la producción científica peruana sobre la COVID-19. Rev Peru Med Exp Salud Publica. 2021 Apr;38(2):224–31.

• Comment 8: plus a wider / global dissemination of biomedical research, including perhaps diseases and conditions that are unique or specific to Peruvian populations...?

Response: Dear Reviewer, thank you for your feedback. We agree with your statement and have incorporated a sentence to that effect: By enhancing these aspects, Peruvian journals can improve research quality and promote global dissemination, with a particular emphasis on conditions unique to Peruvian populations.

• Comment 9: I think you can put the full description here, then add in parenthesis the acronym (to make it clearer for the reader).

Response: Dear Reviewer, thank you for your suggestion. We put the full description of the measures of effect in table 2 as you suggested.

• Comment 10: Is this a repeat of Figure 1? Put one or the other, not both

Response: Dear Reviewer, thank you for your comment. We have addressed it by implementing the suggested change.

---

## [Editor Report · Decision Letter 1]

17 Jun 2024

Statistical characteristics of analytical studies published in Peruvian medical journals from 2021 to 2022: a methodological study

PONE-D-24-15400R1

Dear Dr. Taype-Rondan,

We’re pleased to inform you that your manuscript has been judged scientifically suitable for publication and will be formally accepted for publication once it meets all outstanding technical requirements.

Kind regards,

Alfredo Luis Fort, M.D., M.Sc., Ph.D.

Academic Editor

PLOS ONE

Additional Editor Comments (optional):

Authors have addressed the comments made by reviewers, adding and changing descriptions where necessary. The manuscript is ready for publication, obviously after correcting and addressing any minor spelling or description effects (a couple are addressed in my attached file).

---

## [Editor Report · Acceptance letter]

24 Jun 2024

PONE-D-24-15400R1 

PLOS ONE

Dear Dr. Taype-Rondan, 

I'm pleased to inform you that your manuscript has been deemed suitable for publication in PLOS ONE. Congratulations! Your manuscript is now being handed over to our production team.

Kind regards, 

on behalf of

Dr. Alfredo Luis Fort 

Academic Editor

PLOS ONE